# Flatland and Beyond: Mutual Information Across Geometries

Youssef Wally[1], Johan Mylius-Kroken[1], Michael Kampffmeyer[1,2],
Rezvan Ehsani[3], Vladan Milosevic[3], Elisabeth Wetzer[1]

[1]Department of Physics and Technology, UiT The Arctic University of Norway
[1]Tromsø, Norway
[2] Norwegian Computing Center
[2] Oslo, Norway
[3] Department of Clinical Medicine, University of Bergen
[3] Bergen, Norway

{youssef.m.wally,johan.m.kroken,elisabeth.wetzer,michael.c.kampffmeyer}@uit.no
{v.milosevic,rezvan.ehsani}@uib.no

## Abstract

*Hyperbolic representation learning has shown compelling advantages over conventional Euclidean representation learning in modeling hierarchical relationships in data. In this work, we evaluate its potential to capture biological relations between cell types in highly multiplexed imaging data, where capturing subtle, hierarchical relationships between cell types is crucial to understand tissue composition and functionality. Using a recent and thoroughly validated 42-marker Imaging Mass Cytometry (IMC) dataset of breast cancer tissue, we embed cells into both Euclidean and Lorentzian latent spaces via a fully hyperbolic variational autoencoder. We then introduce an information-theoretic framework based on $k$-nearest neighbor estimators to rigorously quantify the clustering performance in each geometry using mutual information and conditional mutual information. Our results reveal that hyperbolic embeddings retain significantly more biologically relevant information than their Euclidean counterparts. We further provide open-source tools to extend Kraskov-Stögbauer-Grassberger based mutual information estimation to Lorentzian geodesic spaces, and to enable UMAP visualizations with hyperbolic distance metrics. This work contributes a principled evaluation method for geometry-aware learning and supports the growing evidence of hyperbolic geometry's benefits in spatial biology. Code is available at:* https://github.com/youssefwally/FlatlandandBeyond

## 1. Introduction

The encoding of multiscale and hierarchical structure in representation learning has long been a goal in both supervised and unsupervised frameworks. Previous works, such as [36], emphasized the importance of representing and disentangling concepts at different levels of abstraction. Hyperbolic representation learning has recently emerged as a powerful paradigm in geometric deep learning, particularly for tasks involving hierarchical, multiscale, or tree-like structures [3, 24, 31, 36]. Its ability to embed data into spaces of negative curvature has shown substantial benefits across a range of applications, from natural language processing and knowledge graphs to recommender systems and computer vision [20, 27, 38]. A growing body of work has demonstrated that embeddings in hyperbolic spaces can significantly outperform their Euclidean counterparts in tasks such as clustering and classification, particularly with data that exhibits hierarchical relationships [12, 14, 28].

This geometric advantage becomes especially relevant in the context of biological data, where hierarchy is often inherent but difficult to model in traditional Euclidean frameworks. One particularly impactful application is its potential to **capture biological relations** between cell types from multiplexed imaging data, such as Imaging Mass Cytometry (IMC). IMC enables the simultaneous measurement of dozens of protein markers at subcellular resolution through metal-conjugated antibody staining and time-of-flight mass spectrometry. Since different cell types can be characterized by distinct combinations of protein co-expression, identifying these patterns—particularly in an unsupervised setting—is essential for discovering new cellular subtypes and functional states in tissues [21]. The biological systems under investigation are inherently complex and often hierar-

chical, making them a natural candidate for embedding in hyperbolic spaces [29].

Despite the intuitive and visual appeal of hyperbolic embeddings for biological clustering, **rigorously analyzing** that they outperform Euclidean alternatives remains a challenge. Most works rely on qualitative results, with quantitative comparisons often limited or entirely omitted due to the lack of geometry-agnostic evaluation metrics. In this work, we address this gap by proposing an **information-theoretic evaluation framework** to compare the clustering performance of embeddings across geometric spaces, leveraging mutual information (MI) and conditional mutual information (CMI) to quantify the relationship between geometric distance and cell type labels, enabling principled evaluation independent of clustering assumptions. We show that hyperbolic (specifically Lorentzian) embeddings better capture the underlying structure of single-cell IMC data compared to Euclidean embeddings. To support further research in this area, we also release a software package extending the Kraskov-Stögbauer-Grassberger (KSG) MI estimator to operate on Lorentzian geodesic distances, and provide tools to integrate hyperbolic distances into UMAP visualizations.

In Summary, our main contributions are:

- We introduce an information-theoretic evaluation framework for comparing clustering performance across non-Euclidean and Euclidean geometries, based on a generalized kNN-based mutual information estimator.
- We extend the KSG estimator to work with Lorentzian geodesic distances, enabling mutual information estimation in hyperbolic space.
- We demonstrate that hyperbolic latent spaces, specifically Lorentzian variational autoencoders, better capture hierarchical cell taxonomies identified by IMC.
- We provide quantitative and statistical analysis, including McNemar tests and confusion matrix difference visualizations, to show the superiority of hyperbolic embeddings over their Euclidean counterparts.
- We release our implementation, including extensions to UMAP for hyperbolic distance inputs, to support further research in geometric deep learning for spatial biology.

## 2. Related work

### 2.1. Hyperbolic Representation Learning

Hyperbolic spaces, particularly those of constant negative curvature $K < 0$, have emerged as a powerful alternative to Euclidean embedding spaces due to their exponential volume growth, which allows efficient encoding of hierarchical and tree-like data structures [3, 24, 31, 36]. The Poincaré ball model [7, 12, 37], defined as the Riemannian manifold $\mathbb{B}^n = \{x \in \mathbb{R}^n : -K\|x\|^2 < 1\}$ with metric tensor

$$g_x = \left( \frac{2}{1 - \|x\|^2} \right)^2 I_n$$

where $I_n$ is the $n \times n$ identity matrix, $\|x\|$ is the Euclidean norm of $x$, and $g_x$ is a conformal scaling of the Euclidean metric, a factor that depends on the distance from the origin.

This has been widely used in deep learning due to its conformality and closed-form geodesics. More recently, another hypoerbolic model - the Lorentz model [1, 3, 11] - has gained traction due to its better principled optimization frameworks [22]. It embeds points on the upper sheet of the two-sheeted hyperboloid in Minkowski space $\mathbb{R}^{n+1}$, where distances are defined via the Lorentzian scalar product

$$\langle x, y \rangle_L = -x_0 y_0 + \sum_{i=1}^{n} x_i y_i,$$

and geodesic distance is given by

$$d_L(x, y) = |K| \operatorname{arcosh}\left( -\frac{\langle x, y \rangle_L}{|K|} \right).$$

## 3. Methodology

While interest in hyperbolic representation learning continues to grow, quantitatively evaluating clustering performance across different geometries remains an open challenge. Traditional clustering metrics and visualization techniques were designed with Euclidean assumptions, thus often misrepresent the structure and quality of embeddings in hyperbolic spaces. In this work, we tackle this problem by introducing an MI-based, geometry-agnostic methodology.

In the following, we first introduce the Hyperbolic Variational Autoencoder, which enables learning latent representations in Lorentzian space; next, we discuss the limitations of conventional evaluation techniques when applied to non-Euclidean embeddings; finally, we present fully agnostic evaluation metrics, which provide a robust basis for comparing embeddings across geometries without relying on space-specific assumptions.

### 3.1. Hyperbolic Variational Autoencoder

We adopt the fully hyperbolic convolutional neural network (HCNN) introduced by [1] as a recent hyperbolic CNN, benchmarking it against an equivalent Euclidean architecture. Our approach to evaluate their performance to learn representations that capture underlying biological relationships between cells is completely model agnostic, at the cost of requiring class labels for evaluation on the test set. For a controlled comparison, each Euclidean baseline is adapted to the hyperbolic setting by systematically substituting corresponding modules with their hyperbolic counterparts.

The encoder network outputs parameters of a wrapped normal distribution on the hyperboloid $\mathbb{H}^n$, and latent sampling is done using gyrovector analogues of the reparameterization trick. The decoder operates in hyperbolic space using Lorentz-equivariant convolutions to map from $\mathbb{H}^n$ back to Euclidean observation space.

The loss function consists of a hyperbolic reconstruction term. Formally, the evidence lower bound (ELBO) [23] $\mathcal{L}$ is given by

$$\mathcal{L} = \mathbb{E}_{q_\phi(z|x)} \left[ \log p_\theta(x|z) \right],$$

with all distributions and distances defined on $\mathbb{H}^n$.

Variational Autoencoders (VAEs) have been extensively utilized in hyperbolic neural network (HNN) research to model latent representations in non-Euclidean spaces [9, 18, 23, 25]. Prior work has demonstrated that HNNs can produce more expressive embeddings in low-dimensional regimes [1], making them particularly well-suited for integration with VAEs. However, to the best of our knowledge, we introduce the first principled approach for quantitatively comparing Euclidean and hyperbolic latent spaces using standardized metrics.

The authors in [1] provide a HCNN Variational Autoencoder (HCNN-VAE) which we use in this work.

## 3.2. Evaluating Clustering Quality and Visualization Across Geometries

Conventional clustering metrics like the Silhouette Score or Average Distortion Index (ADI) [4, 30, 34] assume Euclidean properties—linear distances, convexity, and isotropic neighborhoods. These assumptions fail in hyperbolic spaces, where distances grow exponentially and neighborhoods are curvature-dependent. Even substituting Euclidean distances with geodesics does not resolve incomparability issues due to the indefinite nature of the inner product and the mismatch with underlying geometric intuitions. Lorentzian inner products are indefinite, leading to undefined or non-interpretable values in some cases. Moreover, many clustering measures rely on uniform notions of "spread" or proximity that are inconsistent with the geometry of curved spaces. As a result, direct metric comparisons between hyperbolic and Euclidean embeddings often fail to capture meaningful structural differences. To overcome this challenge and directly compare if learned representation align better with biological knowledge in a Euclidean or Lorentzian space, we propose to use kNN based mutual information estimation as a geometry-agnostic tool for clustering evaluation. Likewise, visualization tools such as UMAP [19] and t-SNE [35] are formulated under Euclidean assumptions and often introduce artifacts when applied to hyperbolic data.

## 3.3. Fully Agnostic Evaluation Metrics

To circumvent these problems, we adopt a non-parametric information-theoretic and geometry-agnostic estimator for mutual information based on k-nearest neighbors (kNN). Specifically, we use the KSG estimator [17], extended to operate on general metric spaces — including those induced by Lorentzian geodesics. This has several advantages:

**Geometry-Agnostic** MI estimators (e.g., KSG, Local Non-Uniformity Correction or nearest-neighbor entropy estimates) [8, 15, 17] work directly with distances or densities, agnostic to the underlying space. As long as a distance measure is defined, MI estimation proceeds without requiring the geometry to be flat, convex, or isotropic. In particular, we can: *(i)* Define kNNs based on Lorentzian or geodesic distance; *(ii)* Estimate entropies and conditional entropies directly; *(iii)* Evaluate dependency structure between Euclidean and hyperbolic representations. This allows us to quantify how much information one representation preserves about the other, without requiring clustering assumptions or metric preservation [17].

**Local Structure Sensitive** Unlike Silhouette or ADI, which summarize global structure, MI estimation via kNN is local and distribution-sensitive. Because kNN adapts to the local density, it can more robustly compare representations where neighborhood consistency is key — such as checking whether semantic neighbors in Euclidean space remain close in hyperbolic embeddings.

**Alignment Across Representations** Our goal is not merely to assess clustering, but to compare structural fidelity between two spaces. MI offers a principled way to do this: by estimating $I(X;Y)$, where $X$ is the Euclidean representation and $Y$ is the hyperbolic one, we can determine how much information one retains about the other. This is valuable for validating learned representations or dimensionality reduction pipelines between different geometries.

### 3.3.1. Kraskov-Stögbauer-Grassberger (KSG) estimator

Given random variables $X$, $Y$, and a joint sample $\{(x_i, y_i)\}_{i=1}^N$, the KSG estimator is defined as

$$I(X;Y) \approx \psi(k) + \psi(N)$$
$$- \frac{1}{N} \sum_{i=1}^N \left[ \psi(n_x^{(i)} + 1) + \psi(n_y^{(i)} + 1) \right], \quad (1)$$

where $\psi(\cdot)$ is the digamma function, $n_x^{(i)}$ and $n_y^{(i)}$ are the number of neighbors within the $\varepsilon_i$-ball around $x_i$ and $y_i$ respectively, excluding the query point. The radius $\varepsilon_i$ is defined as the maximum distance to the $k$-th nearest neighbor in the joint space.

To adapt KSG for this setting, we introduce a generalized, tensorized implementation that operates on a precomputed 3D pairwise distance array $D \in \mathbb{R}^{p \times N \times N}$, where $p$ is the number of variables, and $D_{v,i,j}$ denotes the distance between samples $i$ and $j$ along variable $v$. Crucially, all distances are computed using the geometry under evaluation: Euclidean norm for baseline model, and Lorentzian geodesics for hyperbolic one. As all terms are computed directly from distances, our evaluation is invariant to coordinate charts and dimensionality, making it a robust tool

for comparing clustering quality across geometric embeddings. Moreover, our implementation supports CMI estimation:

$$I(X;Y|Z) \approx \psi(k) + \psi(N)$$
$$- \frac{1}{N} \sum_{i=1}^{N} \left[ \psi(n_x^{(i)} + 1) + \psi(n_y^{(i)} + 1) - \psi(n_z^{(i)} + 1) \right]$$
(2)

This enables information-theoretic analysis not only of mutual dependence between variables, but also of how that dependence is modulated by auxiliary variables—an important aspect when studying hierarchical structures in Lorentzian spaces.

# 4. Experiments

## 4.1. Datasets

### 4.1.1. IMC

We use the imaging mass cytometry (IMC) dataset introduced in the recent work by [29]. The study presents a meticulously curated 42-marker antibody panel optimized for the phenotypic and spatial characterization of the tumor microenvironment (TME), with a particular emphasis on cancer-associated fibroblasts (CAFs) in breast cancer.

The IMC technique enables simultaneous measurement of protein markers at subcellular resolution by labeling antibodies with metal isotopes and quantifying them via time-of-flight mass spectrometry. In this dataset, tissue sections from breast cancer patients were stained and imaged, producing spatially resolved, single-cell data with high dimensionality and minimal signal overlap—critical for capturing complex phenotypic landscapes within the TME.

The dataset comprises data from 10 patients, with 3 images per patient, resulting in a total of 84,852 single-cell instances. We partitioned the dataset into training, validation, and test subsets using an 80/10/20 stratified split at the cell level: 67,881 cells for training, 6,788 for validation, and 16,971 for testing.

To support multiscale analyses, the dataset includes hierarchical cell type annotations across 4 levels of granularity, however, we only use the first 3. Level 1 categorizes cells into broad functional groups: Cancer cells, Immune cells, Endothelial cells, and Fibroblasts. Level 2 further subdivides Immune cells into finer subtypes, including B cells, T cells, Macrophages (M cells), NK cells, Other immune cells, Pericytes, CD16+, CD3+CD16+, CD3+CD20+, and CD3+CD68+. Level 3 provides an even more detailed view by resolving Macrophages into M0, M1, M2, and MDP, and T cells into MemoryT_CD4+, MemoryT_CD8+, Tcyto (cytotoxic T cells), Th (T helper cells), Treg (regulatory T cells), Other T cells, CD8+CD4+, and CD8-CD4-.

### 4.1.2. MNIST

To complement our analysis on biologically complex data, we also evaluate our method on the widely-used MNIST dataset of handwritten digits in a more controllable environment. MNIST [5] serves as a canonical benchmark in representation learning and clustering tasks, providing a well-understood, high-signal testbed for validating geometry-aware embedding methods.

## 4.2. Implementation Details

All experiments are conducted using PyTorch [26], and the hyperbolic models are trained using adaptive Riemannian optimization techniques [2], as implemented in the Geoopt library [13], with computations carried out in 32-bit floating point precision as in [1].

We evaluate vanilla VAEs on our own dataset **IMC**. Our comparisons include the HCNN-VAE and Euclidean-VAE. Specifically, we evaluate the HCNN models using the wrapped normal distribution in the Lorentz model [23]. Additionally, we evaluated our methods on the MNIST dataset, which exhibited consistent patterns with those observed in the IMC dataset.

The input to the VAE consisted of individual cells segmented from tissue images, with each of the 35 channels representing a constant expression value summarizing the activity of a specific protein marker within that cell. **All analyses are done on the test set.**

## 4.3. Analysis of latent embeddings

To obtain comparable latent representations in both hyperbolic and Euclidean geometries, we train the HCNN-VAE and the Euclidean-VAE independently using reconstruction loss as a common objective. By aligning the training objective across both models, we ensure that any observed differences in representation quality can be attributed to the geometry of the latent space rather than differences in optimization. This setup allows us to derive latent embeddings in hyperbolic and Euclidean spaces under consistent reconstruction constraints, facilitating a fair and meaningful comparison between the two geometries.

The structure of the latent embedding space plays a central role in VAEs, as it determines how input features are encoded and subsequently utilized for generation. To better understand the geometry of the learned representations, we perform a qualitative and quantitative analysis of the latent embeddings inferred by the trained VAEs. Specifically, we encode the test set images and project the resulting embeddings for visualization.

### 4.3.1. Qualitative Analysis

To visualise the structure of the learned latent representations, we extract embeddings from both Euclidean and hyperbolic VAEs, originally 128 space dimensions and 1

time dimension, and project them into two dimensions using UMAP [19]. For each trained model, we pass the test set through the encoder to obtain latent vectors, which are subsequently visualized. While UMAP is a widely used tool for visualizing high-dimensional data, its application to hyperboloid spaces is inherently limited due to its underlying Euclidean assumptions. Specifically, standard UMAP operates under the premise that input data resides in a Euclidean space and relies on Euclidean distance metrics to construct neighborhood graphs. This poses a mismatch when applied directly to embeddings in the Lorentz model, where distances are defined with respect to a pseudo-Riemannian geometry. As a result, applying UMAP to Lorentzian embeddings can distort the geometric relationships among points, leading to misleading visualizations. Unlike previous work, to address this issue, we manually compute normalised pairwise Lorentzian distances between latent embeddings and leverage UMAP's precomputed distance mode, allowing us to preserve the intrinsic geometry of the hyperbolic space during dimensionality reduction.

### 4.3.2. Quantitative Analysis

While qualitative analysis through embedding visualizations provides intuitive insights into structural differences between Euclidean and Lorentzian spaces, it remains inherently subjective and limited by human interpretation. In contrast, quantitative analysis offers a robust and objective framework for evaluating representational quality, using well-defined statistical and geometric measures. These metrics enable reproducible, data-driven comparisons that can generalize across datasets and experimental settings.

To compute MI and CMI using the kNN approach, we rely on identifying local neighborhoods for each observation in the dataset based on a user-defined distance metric. For a given point, we calculate the $\ell^\infty$ (maximum coordinate-wise) distance to all other points across the relevant subset of variables $v$, whether from $X$, $Y$, or $Z$, using a precomputed pairwise distance array. The $k$-th smallest such distance defines a radius $\rho$ around the point, which is then used to count the number of neighbors lying within this radius in the joint space ($XYZ$), and in the marginal subspaces ($XZ$, $YZ$, and $Z$). These counts are then plugged into the previous mentioned equations $I(X;Y)$ and $I(X;Y \mid Z)$. Importantly, this procedure is fully agnostic to the ambient geometry because it relies only on the distance matrix: we can define neighborhoods using Euclidean distances, Lorentzian distances, or any other valid metric, allowing for a direct comparison of information content across different geometrical representations.

To incorporate categorical variables, such as class labels, into the MI and CMI estimation framework, we define a discrete distance metric that assigns a fixed, nonzero distance when two values differ and zero when they are equal. Specifically, for any categorical variable, we construct a pairwise distance matrix where each entry is either 0 (if the two observations belong to the same category) or a fixed scalar (typically 1) if they differ. This binary distance captures label similarity without imposing an artificial ordering or metric structure on the categories. When computing neighborhoods for MI or CMI estimation, these categorical distances are treated on the same footing as continuous distances: they contribute to the $\ell^\infty$ distance used to determine the local radius $\rho$ for kNN counting. This approach allows categorical variables to be integrated naturally into the neighborhood-based estimation framework while preserving their non-metric nature, ensuring accurate and geometry-agnostic information estimates. Therefore, the distances fed should be normalised between 0 and the assigned nonzero distance.

Finally, by reusing the kNN structure, we can derive empirical neighborhood-based confusion matrices, enabling classification-based evaluations that are intrinsically aligned with the MI estimation process.

## 5. Results

### 5.1. Qualitative Results

A visual comparison of embeddings in Lorentzian and Euclidean spaces computed using neighborhood graphs constructed based on geodesic distances or euclidean distances depending on the space geometry, as seen in Fig. 1, reveals notable structural differences that underscore the representational advantages of hyperbolic geometry. In Lorentz space, clusters are consistently more compact, reflecting the space's exponential volume growth. This is particularly evident in the IMC dataset, where the minority class of Endothelial Cells —comprising only 8.40% of the total data—is more distinctly and tightly clustered in Lorentzian embeddings, whereas it is more diffuse and harder to separate in Euclidean space. This suggests that the Lorentzian geometry better captures subtle, semantically meaningful differences even for underrepresented classes.

Additionally, in the MNIST dataset, we observe that certain instances of the digit "3" occupy a "higher level" in the Lorentzian embedding space—appearing scattered above clusters corresponding to digits "0", "6", and "8". This behavior highlights Lorentz space's capacity to encode semantic ambiguity: some "3"s share more visual similarity with these digits, and hyperbolic embeddings naturally reflect this by positioning them in intermediary regions, in contrast to the flatter separation seen in Euclidean projections. Moreover, we observe that MNIST Lorentzian embeddings are closer to the boundary since they follow a less hierarchal structure.

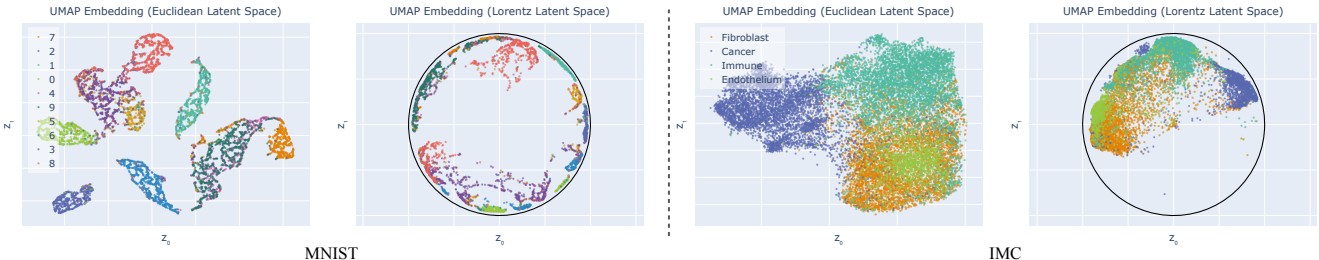

Figure 1. Embeddings in 2D latent space of VAEs. Colors represent ground truth labels.

Table 1. Estimated MI and CMI on IMC and MNIST test sets.

| Quantity | IMC | MNIST |
|---|---|---|
| $MI(D_L; C)$ | **1.07** | **1.86** |
| $MI(D_E; C)$ | 0.96 | 1.78 |
| $MI(D_L; D_E)$ | 0.01 | 4.03 |
| $CMI(D_L; C \mid D_E)$ | **1.06** | **0.16** |
| $CMI(D_E; C \mid D_L)$ | 0.00 | 0.09 |

## 5.2. Quantitative Results

We quantitatively assess how each geometry—Euclidean and Lorentzian—captures class-relevant structure in the embedding space. Specifically, we estimate the MI between the Lorentzian distance matrix and the class labels, $I(D_L; C)$, and between the Euclidean distance matrix and the class labels, $I(D_E; C)$, where $D_L$ and $D_E$ are the pairwise distance matrices computed under Lorentzian and Euclidean metrics, respectively, and $C$ denotes the discrete class labels. Additionally, we compute the MI between the two distance matrices themselves, $I(D_L; D_E)$, to understand how much geometric information is shared between the two representations. To further probe the complementarity of the geometries, we compute two conditional MI quantities: $I(D_L; C \mid D_E)$, which captures the incremental information that Lorentz geometry provides about the labels beyond what is already explained by Euclidean structure, and $I(D_E; C \mid D_L)$, which evaluates the converse. These quantities provide a more nuanced, information-theoretic view of the expressive power of each space, revealing not just which geometry encodes class information more efficiently, but also whether either provides non-redundant signal beyond the other. The results are shown in Table 1.

The MI results quantitatively confirm the qualitative observations. The MI between Lorentzian distances and class labels, $I(D_L; C) = 1.07$, is considerably higher than the Euclidean counterpart, $I(D_E; C) = 0.96$, indicating that the Lorentzian geometry encodes more class-relevant information. This aligns closely with visual observations—particularly the greater compactness of clusters

and improved separation of minority classes like the endothelial cells. Furthermore, the MI between Lorentzian and Euclidean distances is negligible, $I(D_L; D_E) = 0.01$, suggesting that the two geometries capture fundamentally different structural aspects of the data rather than being redundant encodings. The CMI results reinforce this interpretation: conditioning on Euclidean distances does not significantly reduce the MI between Lorentzian distances and class labels, $I(D_L; C \mid D_E) = 1.06$, while the reverse $I(D_E; C \mid D_L) = 0.00$ indicates that Euclidean distances contribute no additional class information beyond what is already captured by the Lorentzian geometry. Together, these results underscore the greater expressive power and discriminative capacity of the Lorentzian space, providing a robust theoretical foundation for its observed empirical advantages in embedding structure and cluster quality.

Furthermore, we can leverage the pairwise distance matrices—computed using the appropriate geometry—to train a k-nearest neighbors (kNN) classifier. This enables us to evaluate the clustering quality that is agnostic to the embedding space, allowing for a direct comparison between hyperbolic and Euclidean representations. Moreover, because kNN relies solely on local neighborhood structure without learning additional parameters, this evaluation isolates the quality of the embeddings themselves, avoiding confounding effects from downstream model capacity. By applying the same classification framework across geometries, we obtain comparable accuracy scores, adjusted rand indices (ARI) [10], and confusion matrices that further quantify the discriminative structure captured in each latent space.

The confusion matrices and derived metrics, shown in Fig. 2, support the claim that kNN-based MI estimation effectively captures the alignment between local distance distributions and class labels. When evaluating the kNN classifier on the Euclidean embeddings of the test set, the overall classification performance is moderate, with an accuracy of 81.01% on level 1 and 64.59% on level 3 and an ARI of 0.565 and 0.505, respectively. Most notably, the classifier struggles significantly with the minority Endothelial Cells, correctly identifying only 34% of instances, with 56% misclassified as Fibroblasts. In contrast, when

the same classifier is trained and evaluated using distances from the Lorentzian space, we observe substantial improvements. The accuracy increases to 86.54% on level 1 and 70.11% on level 3 and the ARI to 0.687 on level 1 and 0.637 on level 3. The most dramatic enhancement is again seen in the minority Endothelial Cells, with correct identification improving to 51% and misclassification reducing to 41%. This aligns well with our qualitative observations where the Lorentzian embeddings produced more compact and geometrically meaningful clusters, especially benefiting underrepresented categories like Endothelial Cells. The Lorentzian space appears to better capture the subtle relational structure of the data, which allows the classifier to make more confident and accurate decisions even with sparse class examples.

Further results and statistical tests can be found in the supplementary material.

## 6. Discussion

Our findings demonstrate that hyperbolic embedding models, particularly those trained in Lorentzian space, are not only effective at capturing clustering structure in high-dimensional biological data but also align remarkably well with current medical understanding of cell identities and phenotypic transitions. The use of Lorentzian geometry allows for a more natural encoding of biological hierarchies, as reflected in the qualitative inspection of model outputs and confirmed by domain experts in translation medicine and IMC imaging. Crucially, these insights are enabled by our proposed information-theoretic evaluation framework, which allows for principled, geometry-agnostic comparison of clustering performance and reveals the superiority of hyperbolic embeddings that would otherwise be obscured by traditional Euclidean-based metrics.

Several apparent misclassifications in our results are in fact biologically meaningful. Both cell types express high levels of vimentin, particularly in regions where they are spatially co-located, making them difficult to distinguish with current marker panels. Compounding this, CD34 is also known to be expressed in some type of cancer associated fibroblasts [6, 33], which can result in both classification ambiguity and protein signal spillover. The model's tendency to conflate endothelial cells and fibroblast populations is thus not a failure of representation but rather a reflection of biological and technical uncertainty.

A similar observation holds for immune cell types. Immune subtypes such as T cells, NK cells, and macrophages are known to express overlapping markers in varying degrees, and they often occupy shared tissue niches, contributing to lateral signal spillover. For instance, CD16 is a marker present in both NK cells and monocytes, while granzyme B expression — necessary for defining NK cells — can be minimal. These subtleties often challenge both

manual annotation and automated clustering. Our model reflects this ambiguity, but crucially, the Lorentzian version tends to organize immune subtypes according to functional and developmental relationships, rather than merely spatial proximity or surface-level marker expression. Notably, Memory CD8+ T cells are embedded more closely to other T cell types, rather than being erroneously pulled toward CD8+CD4+ double-positive regions [32].

In fact, one of the most striking advantages of hyperbolic embeddings lies in their capacity to preserve such hierarchical relationships. Unlike Euclidean embeddings, which impose an isotropic geometry ill-suited for tree-like or taxonomic structures, hyperbolic spaces naturally accommodate such topologies. The Lorentzian model, with its capacity to represent indefinite inner products and exponential volume growth, models the nested and branching lineage structure of immune and stromal compartments more accurately. For example, ambiguous macrophage states such as M0 and MDP are embedded near M1 and M2 macrophages, not with unrelated fibroblast populations — also a misclassification more commonly observed in Euclidean space. This structural consistency is particularly important for downstream tasks such as subtype discovery or treatment response modeling, where the preservation of biological topology directly impacts interpretability.

We also observed improved modeling of fibroblast heterogeneity, an area of growing importance in tumor microenvironment research. Fibroblast subtypes, including cancer-associated fibroblasts (CAFs) and pericytes, are notoriously hard to distinguish with current marker panels. In our dataset, pericyte identification was hindered by equivocal staining of MCAM, a known limitation acknowledged by the original study [29]. Despite this, the Lorentzian model managed to preserve a degree of distinction between fibroblast-related types and other lineages, while still allowing for fluid boundaries in the embedding space. This is crucial for modeling uncertain or transitional phenotypes without forcing hard, discrete classifications where they may not exist biologically.

Overall, while both models inevitably produce some misclassifications due to the complexity and noise inherent in biological data, the Lorentzian model consistently makes misclassifications that are mostly medically interpretable and aligned with current biological understanding. Its embedding space captures transitional phenotypes, lineage proximities, and marker expression overlaps in ways that reflect the known fluidity and hierarchy of cell states. In contrast, the Euclidean model tends to impose artificial boundaries and incorrectly groups unrelated cell types, failing to account for the nuanced relationships that define biological systems. This highlights the critical importance of choosing a geometric representation that reflects the latent structure of the data: in our case, the Lorentzian hyper-

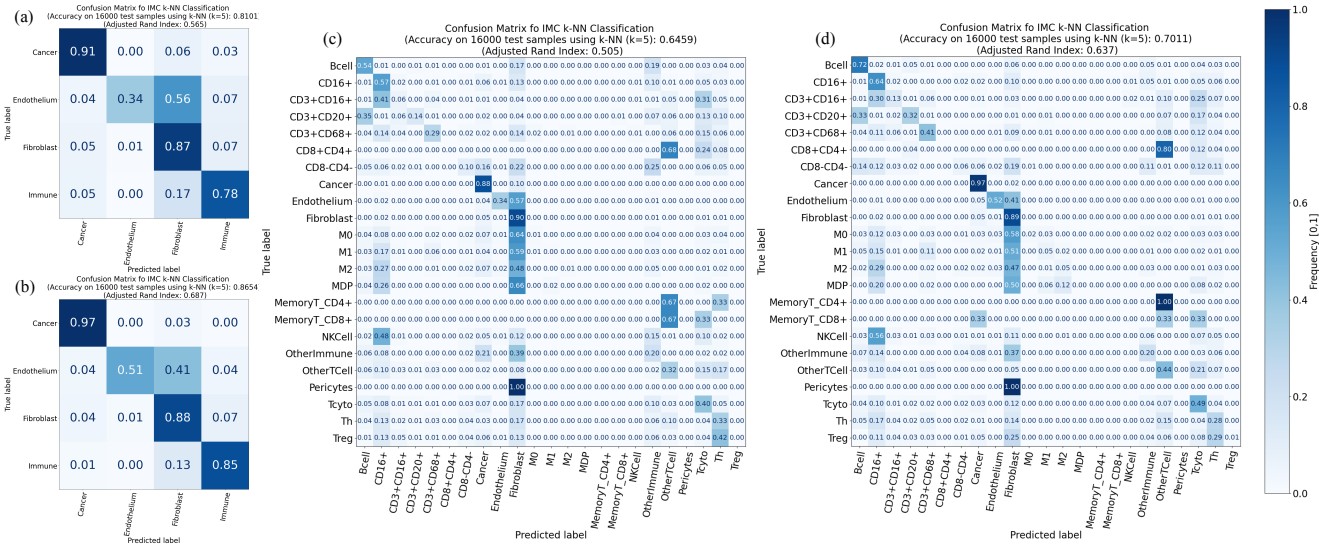

Figure 2. k-nearest neighbor (kNN) classifier results on the IMC test set. (a) Euclidean representations with labels at level 1, (b) Lorentz representations with labels at level 1, (c) Euclidean representations with labels at level 3, and (d) Lorentz representations with labels at level 3

bolic model not only improves clustering metrics, but more importantly, produces representations that are semantically meaningful to domain experts.

## 6.1. Limitations

Although Lorentzian geometry appears to model hierarchical cell type relationships more faithfully, we rely on ground-truth class labels in the test set to estimate MI. This dependence constrains the applicability of the approach to labeled datasets and may limit its generalizability in unsupervised or real-world clinical settings where annotations are scarce or noisy.

Another key limitation is our reliance on downstream visual inspection and qualitative feedback from domain experts to validate this structure. A full biological validation—e.g., through targeted experimental follow-up—is beyond the scope of this study and remains an important direction for future work.

Additionally, while pairwise distance matrices provide interpretable and geometry-consistent basis for downstream tasks such as kNN classification, computing them becomes computationally intensive as dataset size grows. For larger-scale applications, efficient approximation or sampling strategies would be necessary to maintain scalability.

## 7. Conclusions and Outlook

In this work, we used information-theoretic measures to quantify the clustering of unsupervised representation learning techniques across geometric spaces equipped with non-equivalent metrics. Specifically, we investigated the

suitability of the KSG formulation of MI, which is based on kNN statistics and does not assume any specific geometric structure. This allowed for a metric-agnostic comparison of learned latent spaces.

Our qualitative analysis showed that Lorentzian embeddings yield more compact and semantically coherent clusters, especially benefiting minority classes such as Endothelial Cells in the IMC dataset. Quantitative results reinforced these insights: MI and conditional MI revealed that Lorentzian distances carry substantially more information about class labels than their Euclidean counterparts, even when controlling for geometric overlap. Additionally, confusion matrices from kNN classifiers demonstrated that Lorentzian embeddings improve classification accuracy and clustering robustness, particularly for ambiguous or underrepresented samples.

To do so, we extended the classical [16] MI estimator into a geometry-agnostic framework that operates on distance matrices, enabling fair comparisons across different latent geometries. This formulation also allowed the integration of KNN-based classification to derive interpretable metrics like confusion matrices, accuracy, and ARI. Taken together, our findings provide strong evidence that hyperbolic (Lorentzian) geometry offers a more expressive and discriminative embedding space for high-dimensional, structured data, especially in regimes of class minority and subtle feature variations.

## 8. Acknowledgements

This work was supported by the Research Council of Norway, Integreat – Norwegian Centre for knowledge-driven machine learning (project number 332645), FRIPRO grant no. 315029, and the Norwegian Cancer Society (Kreftforeningen).

Special thanks to Arne Östman for his valuable input in discussions and his expertise in cell biology. The authors wish to thank Randi Hope Larvik and Bendik Nordanger for their excellent technical assistance, as well as the Flow Cytometry Core Facility, University of Bergen. The authors would also like to thank Lars Akslen for providing the tissue material.

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
