# Flatland and Beyond: Mutual Information Across Geometries

Youssef Wally[1], Johan Mylius-Kroken[1], Michael Kampffmeyer[1,2],
Rezvan Ehsani[3], Vladan Milosevic[3], Elisabeth Wetzer[1]

[1]Department of Physics and Technology, UiT The Arctic University of Norway
Tromsø, Norway

[2] Norwegian Computing Center
Oslo, Norway

[3] Department of Clinical Medicine, University of Bergen
Bergen, Norway

{youssef.m.wally,johan.m.kroken,elisabeth.wetzer,michael.c.kampffmeyer}@uit.no

{v.milosevic,rezvan.ehsani}@uib.no

## A. Confusion Matices for IMC at Level 2

Fig. 1 shows the corresponding confusion matrix (CM) for IMC level 2 which shows similar patterns as the confusion matrices for classification on level 1 and level 3 as shown in 2.

## B. Comparative Analysis via Confusion Matrix Differencing

To directly assess the qualitative and quantitative differences between clustering outcomes in Euclidean and Lorentzian geometries, we compute and visualize the element-wise difference between their respective confusion matrices. This subtraction highlights where one geometry assigns more samples to a particular predicted class compared to the other.

By visualising the matrix $\Delta$ = CM{Euclidean} − CM{Lorentz}, in Figures 2, 3, and 4, we reveal systematic shifts in classification behavior. Positive entries in $\Delta$ indicate class assignments more prevalent under the Euclidean model, while negative entries highlight where the Lorentz embedding dominates. This comparative visualization allows us to pinpoint the specific cell subtypes and transitions where hyperbolic geometry provides more biologically plausible predictions, and where Euclidean embeddings may introduce misclassifications due to their inability to accommodate hierarchical relationships. The difference matrix thus serves as an interpretable tool to summarize performance gains in a fine-grained, label-wise fashion.

## C. Statistical Comparison via McNemar's Test

To assess whether the differences in classification performance between the Euclidean and Lorentzian models are statistically significant, we apply McNemar's test to their respective predictions. This non-parametric test is designed for paired nominal data and is well-suited for evaluating whether two classifiers differ in accuracy on a per-sample basis, particularly when their predictions are dependent or made on the same test set.

We construct a $2 \times 2$ contingency table, Fig. 5, capturing the number of instances that were (i) correctly classified by both models, (ii) only by the Lorentzian model, (iii) only by the Euclidean model, and (iv) misclassified by both. The McNemar statistic then tests whether the off-diagonal elements — the disagreements between the models — are symmetric. A significant result (typically $p < 0.05$) indicates that the improvement of one model over the other is unlikely to be due to chance. This test thus provides rigorous statistical backing to our claim that the Lorentzian geometry leads to more consistent and biologically meaningful predictions.

Our McNemar test yields a p-value that is orders of magnitude smaller than the commonly used threshold of 0.05, indicating that the null hypothesis—that the Euclidean and Lorentzian models have the same proportion of correct predictions—is strongly rejected. This result is further reinforced by the very large corresponding chi-squared statistic, which quantifies the degree of disagreement between the two models' predictions. The magnitude of this test statistic clearly highlights a substantial and statistically significant difference in predictive behavior between the geometries, underscoring the robustness of the Lorentzian model's improvements in clustering and classification performance.

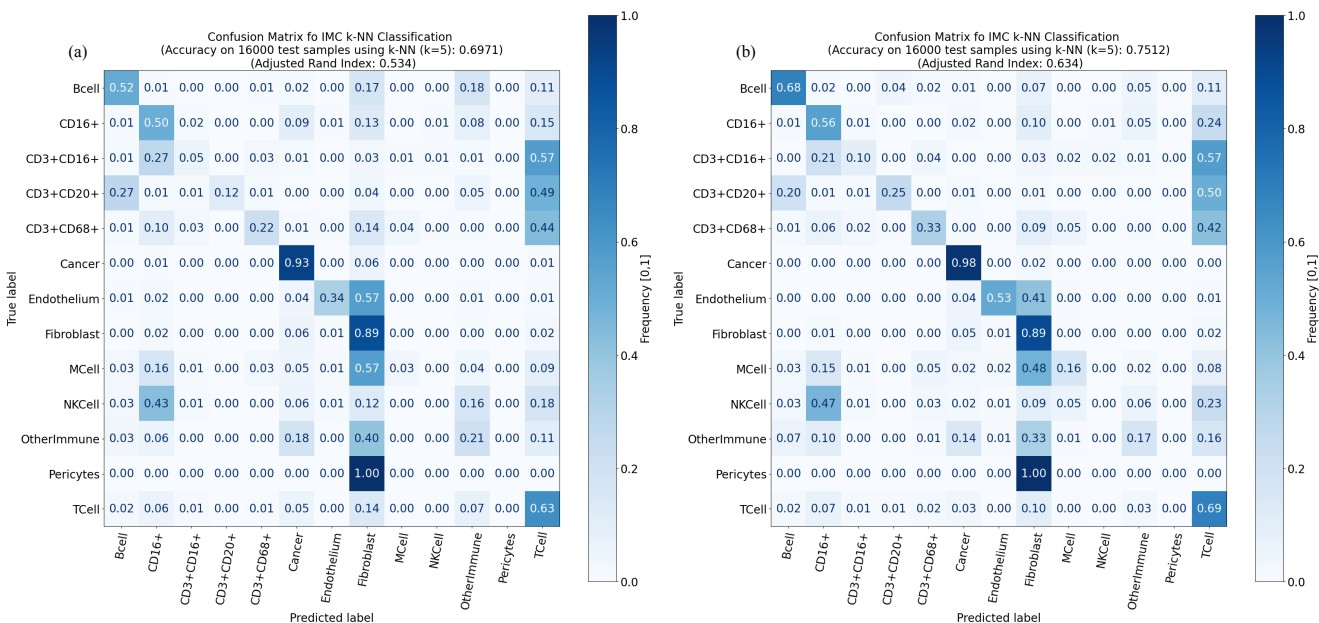

Figure 1. k-nearest neighbor (kNN) classifier results on the IMC testset. (a) Euclidean representation with labels at level 2, (b) Lorentz representation with labels at level 2

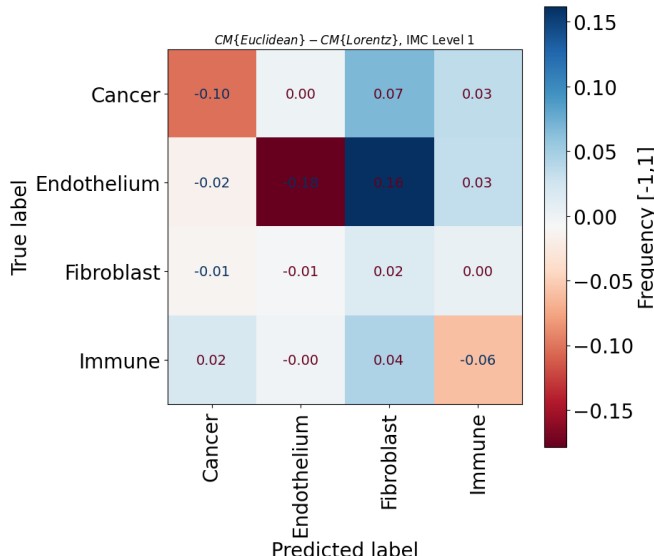

Figure 2. IMC level 1 $\Delta = \mathrm{CM\{Euclidean\}} - \mathrm{CM\{Lorentz\}}$

## D. IMC Markers

Markers used span fibroblast heterogeneity, immune subsets, pancytokeratin and E-Cadherin markers, and functional states (e.g., Ki-67 for proliferation, cleaved caspase 3 as an apoptosis marker). enabling fine-grained dissection of both structural and functional cell phenotypes, facilitating a comprehensive view of CAF niches and their interactions with other cell types. Biomarkers used:

1. ALDH1
2. gamma-catenin (not used in training)
3. MCAM
4. aSMA

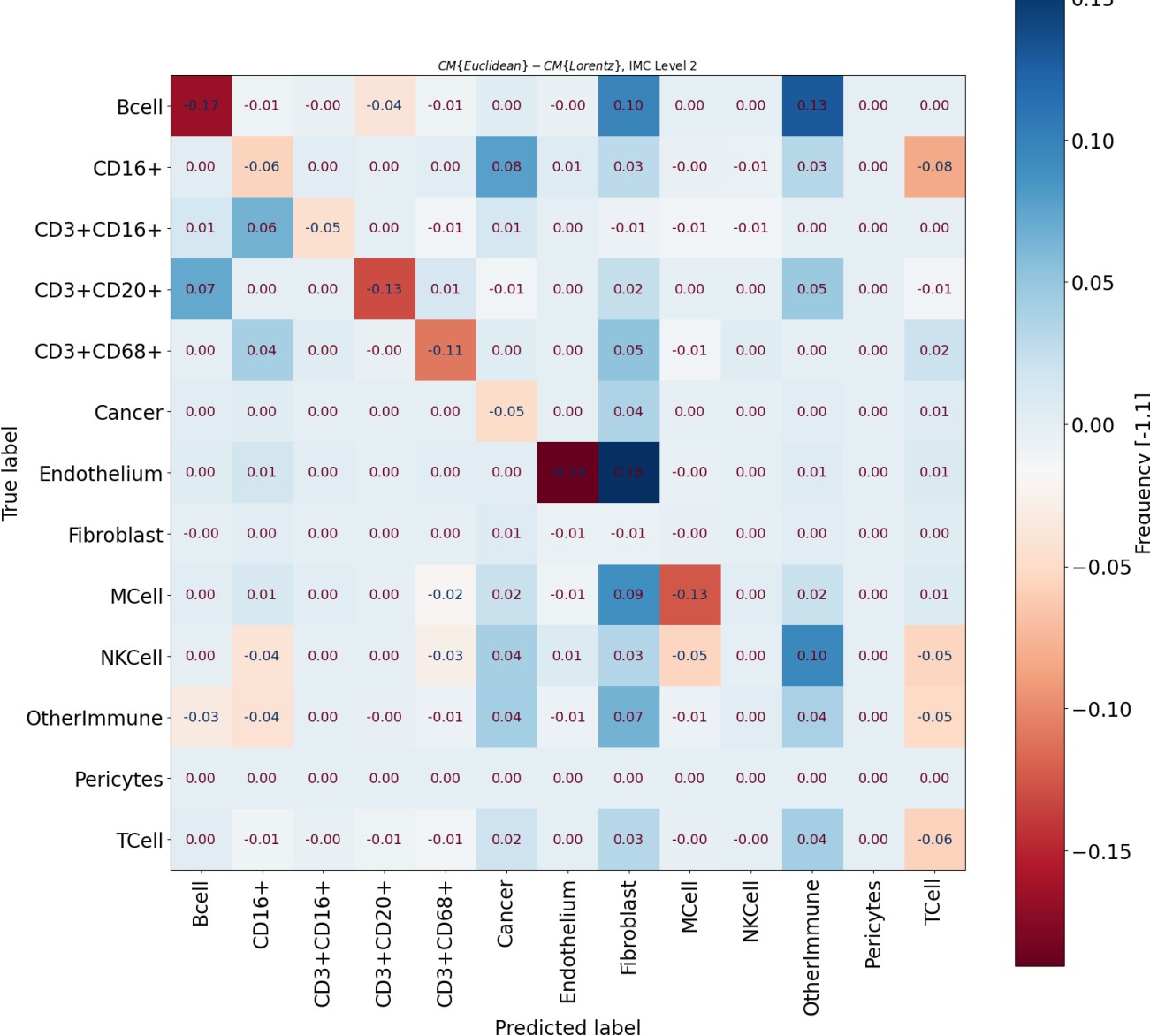

Figure 3. IMC level 2 $\Delta = \mathrm{CM\{Euclidean\}} - \mathrm{CM\{Lorentz\}}$

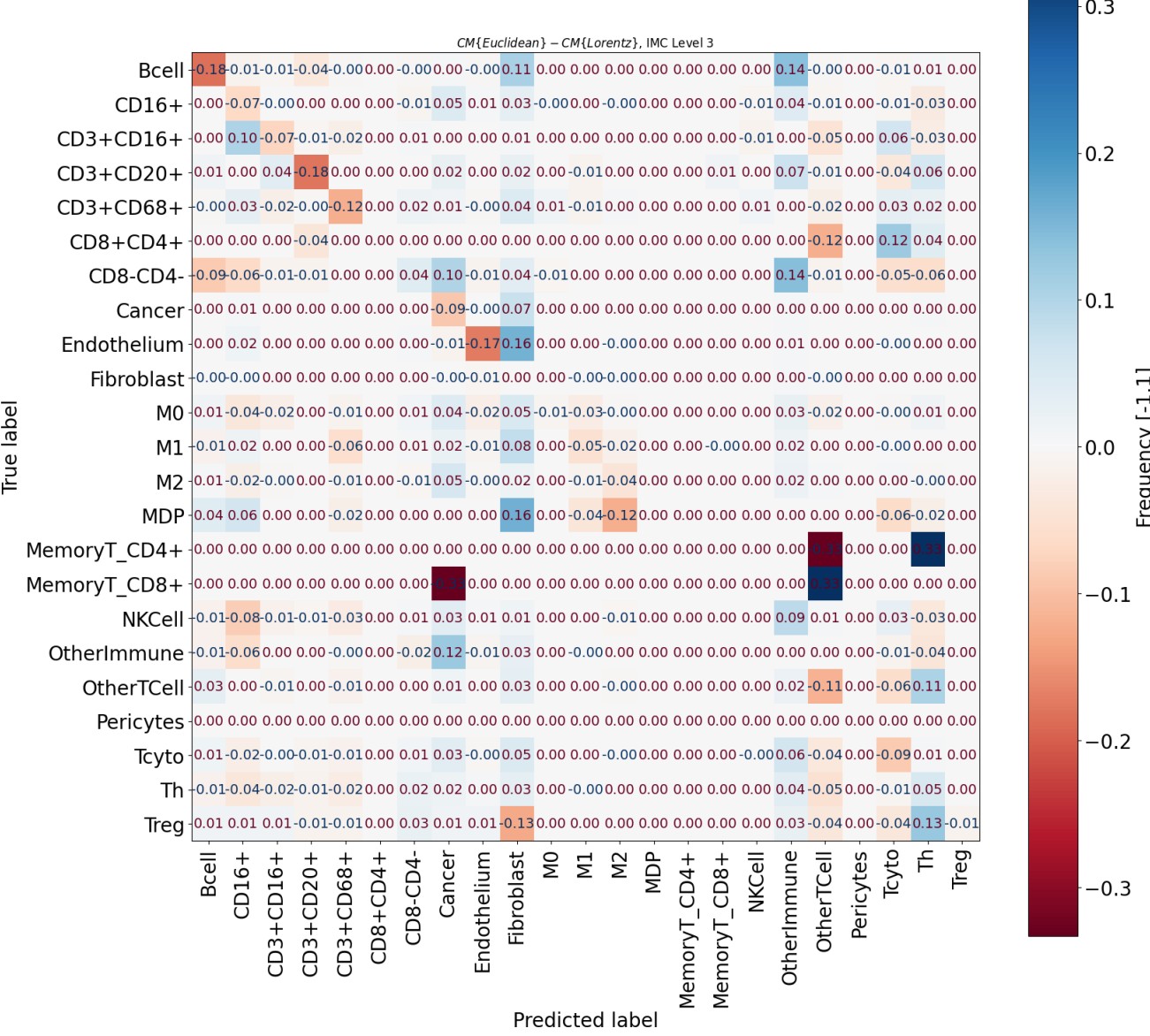

Figure 4. IMC level 3 $\Delta = \mathrm{CM\{Euclidean\}} - \mathrm{CM\{Lorentz\}}$

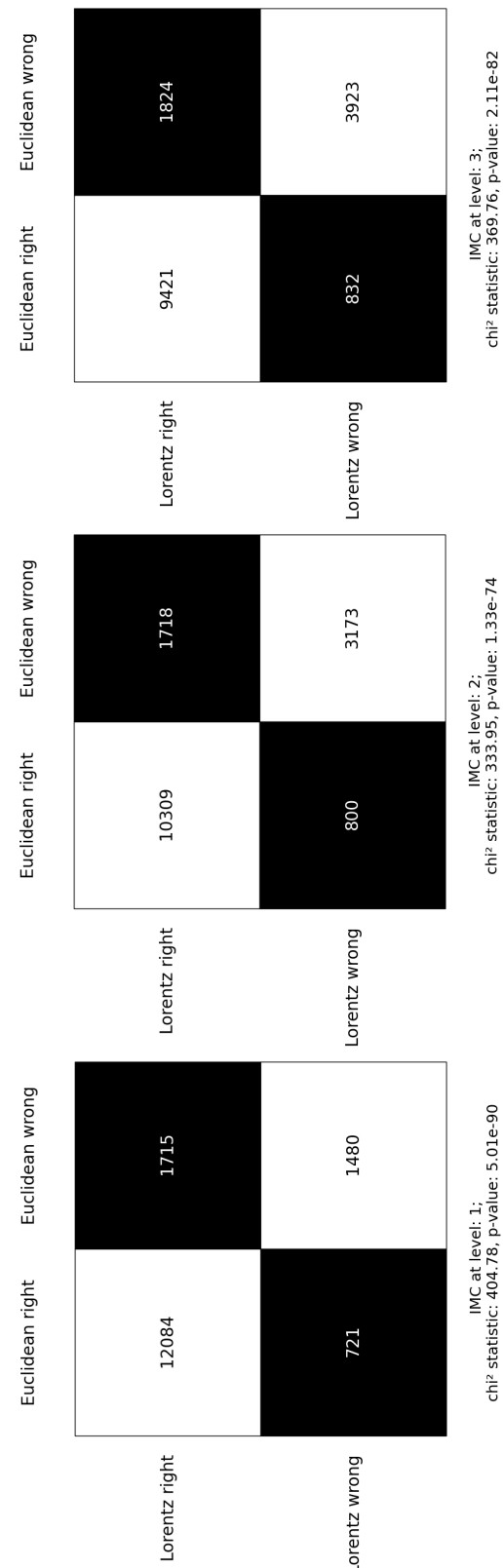

Figure 5. McNemar test between the Euclidean model and Lorentz model predictions.