# OpenReview forum: "Flatland and Beyond: Mutual Information Across Geometries"
_thecvf.com/ICCV/2025/Workshop/BEW — BEW 2025 Oral_

### Official Review · Reviewer_KZmU · 2025-06-30
**hyperbolic embeddings retain significantly more biologically information than Euclidean counterparts**

**Rating:** 4
**Confidence:** 2

**Review:**

This paper employs Hyperbolic embeddings for inferering significantly biological information, which was doing better than Euclidean counterparts. I am not an expert in this area. Mutual information metric was employed in this work. Interesting results are shown in the paper. For example, Figure 1 shows that the Lorentz approach provide interesting clustering results compared to the Euclidean approach. The classification results shown in Figure 2 also support the hyperthesis.

---

### Official Review · Reviewer_kG4J · 2025-07-06
**A solid analysis of hyperbolic embeddings for biological data and valuable metrics**

**Rating:** 5
**Confidence:** 4

**Review:**

This paper presents a comparative analysis of the Euclidean and hyperbolic embeddings learnt by a CNN-based VAE, benchmarked on a domain-specific biological dataset (ICM) and a standard general-purpose dataset (MNIST).

The manuscript is well-structured and pleasant to follow. The method is thoroughly explained and clearly torn down. I appreciate that the proposed method has been assessed on both a domain-specific benchmark and a general-purpose, widely used one, as this choice strengthens the analysis and makes the empirical observations more grounded. I also particularly appreciate the discussion section, which gives an in-depth biological interpretation of the spatial distribution of the learnt embeddings, guiding the reader throughout the technical biological terminology without falling into a murky dissertation.

There are some minor flaws to address. Section 2 provides a quite limited literature background on representation learning, both in Euclidean and hyperbolic spaces, and on existing state-of-the-art techniques; I would recommend expanding this section. Additionally, while carefully commented on throughout Section 5.2, Table 1 could be improved, incorporating some more information regarding how to read it in the caption, so as to let the reader get an impression of the reported results even without going through the section.

Overall, I deem this work valuable and well-targeted for the workshop, and thus I recommend unconditional acceptance.

---

### Official Review · Reviewer_xVsK · 2025-07-08
**Review of Flatland and Beyond: Mutual Information Across Geometries**

**Rating:** 5
**Confidence:** 3

**Review:**

## Summary

This paper proposes a geometry-agnostic framework for evaluating latent representations in hyperbolic versus Euclidean spaces, focusing on biological single-cell imaging data (IMC). Using mutual information (MI) and conditional mutual information (CMI) estimators adapted to Lorentzian geodesic distances, the authors quantify how well embeddings preserve biologically meaningful structures. Experiments on a 42-marker breast cancer IMC dataset and MNIST show that hyperbolic embeddings learned via a fully hyperbolic VAE capture significantly more class-relevant information than Euclidean embeddings, with particular benefits for minority cell types.

## Strengths

- **Geometry-agnostic MI estimation:** The use of the KSG estimator to compute mutual information offers a rigorous, geometry-agnostic metric for comparing embeddings. The quantitative results in Table 1 and the analysis in Sec. 5.2 (lines 473–476) clearly show that Lorentzian embeddings retain more class-relevant information (I(DL; C) = 1.07) than Euclidean embeddings (I(DE; C) = 0.96).

- **Evaluation of hyperbolic embeddings with CMI:** The conditional mutual information experiments (lines 485–490) are particularly insightful. They show that conditioning on Euclidean distances does not diminish the MI between Lorentzian distances and class labels (I(DL; C | DE) = 1.06), while Euclidean distances add no additional information once Lorentzian embeddings are known.

- **Distinct structural encoding:** The observation that MI between Lorentzian and Euclidean distances is negligible (I(DL; DE) = 0.01, lines 481–484) suggests the two geometries capture fundamentally different structural aspects of the data rather than redundant encodings.

- Adapting UMAP with precomputed Lorentzian distances is a practical step given standard UMAP’s Euclidean assumptions.

## Weaknesses

- **Limited discussion on limitations and failure cases:** The paper does not sufficiently clarify under what conditions the proposed method might fail or produce misleading results. For instance, could noise or highly overlapping classes obscure MI estimates?

- **Scope of CMI limited to class labels:** The CMI experiments are only conducted on discrete class labels. It is unclear how this framework would extend to other tasks such as segmentation masks, multi-label classification, bounding boxes, or dense pixel-wise labels, where the “label” may be structured or high-dimensional. I suggest the authors to add a discussion about the extensions in the final version.

- **Correlation of metrics with downstream performance:**  It remains unknown how the MI or CMI metrics correlate with actual downstream task performance beyond classification, e.g., in segmentation [GhadimiAtigh et.al., CVPR 2022] or multi-modal models like HyCoCLIP [Pal et. ICLR 2025].

- **Single network architecture evaluated:** The study focuses exclusively on hyperbolic VAEs (specifically the HCNN-VAE). It would be valuable to test whether the proposed evaluation framework generalizes to other representation learning methods, such as hyperbolic graph neural networks [20, 38], hyperbolic metric learning [12, 37], or hyperbolic embeddings learned via contrastive learning.

- **t-SNE alternatives:** While the paper notes that t-SNE is Euclidean, it overlooks more recent alternatives like CO-SNE [Guo et al., CVPR 2022], which was explicitly designed for hyperbolic data visualization. It would be insightful to plot CO-SNE in Figure 1 and provide a discussion of how the proposed modified UMAP compares with CO-SNE in preserving hyperbolic structure.

## Justification

Despite these weaknesses, this is an interesting paper and provide new insights into comparing euclidean and hyperbolic embedding spaces. It provides valuable methodological contributions and empirical insights, for geometry-agnostic representation learning. Overall, paper seems a good fit for the BEW workshop.

---

### Decision · Program_Chairs · 2025-07-09

**Decision:**

Accept (Oral)

**Comment:**

The majority of the reviews agree towards accepting the paper.  The authors should do their best to address the comments of the reviewers in their final version.  The oral presentations would also present a poster.